# Public support for global vaccine sharing in the COVID-19 pandemic: Evidence from Germany

Ferdinand Geissler[1]*, Felix Hartmann[1], Macartan Humphreys[2,3], Heike Klüver[1], Johannes Giesecke[1]

**1** Humboldt-Universität zu Berlin, Berlin, Germany, **2** WZB Berlin Social Science Center, Berlin, Germany, **3** Columbia University, New York City, New York, United States of America

* ferdinand.geissler@hu-berlin.de

**Data Availability Statement:** Data and replication material is available at https://wzb-ipi.github.io/vaccine_solidarity/.

## Abstract

By September 2021 an estimated 32% of the global population was fully vaccinated for COVID-19 but the global distribution of vaccines was extremely unequal, with 72% or more vaccinated in the ten countries with the highest vaccination rates and less than 2% in the ten countries with the lowest vaccination rates. Given that governments need to secure public support for investments in global vaccine sharing, it is important to understand the levels and drivers of public support for international vaccine solidarity. Using a factorial experiment administered to more than 10,000 online survey respondents in Germany in 2021, we demonstrate that the majority of German citizens are against global inequalities in vaccine distribution. Respondents are supportive of substantive funding amounts, on the order of the most generous contributions provided to date, though still below amounts that are likely needed for a successful global campaign. Public preferences appear largely to be driven by intrinsic concern for the welfare of global populations though are in part explained by material considerations—particularly risks of continued health threats from a failure to vaccinate globally. Strategic considerations are of more limited importance in shaping public opinion; in particular we see no evidence for free riding on contributions by other states. Finally, drawing on an additional survey experiment, we show that there is scope to use information campaigns highlighting international health externalities to augment public support for global campaigns.

## Introduction

Vaccination is the key to overcome the COVID-19 pandemic. Multiple vaccines have been developed in record time and more than 12 billion doses administered and 63% of the global population fully vaccinated by October 2022. Already by September 2021 an estimated 32% of the global population was fully vaccinated. However, the global distribution of the COVID vaccines has been extremely unequal [1]: By that time about 72% of citizens were vaccinated in the ten countries leading in the COVID vaccination campaign, less than 2% of the population

**Funding:** Research for this contribution is part of the Cluster of Excellence "Contestations of the Liberal Script" (EXC 2055, Project-ID: 390715649; https://www.scripts-berlin.eu/), funded by the Deutsche Forschungsgemeinschaft (DFG, German Research Foundation) under Germany´s Excellence Strategy. The funders had no role in study design, data collection and analysis, decision to publish, or preparation of the manuscript.

**Competing interests:** The authors have declared that no competing interests exist.

is vaccinated in the ten most poorly performing countries. Those top 10 and bottom 10 countries are from among 102 countries with populations >500, 000 that report vaccination rates. Strikingly, based on calculations on data by [1], per capita GDP alone explains as much as 56% of this variation in vaccination rates; COVID mortality about 8% (calculations available in replication materials).

Besides the evident inequities, and the economic and health threats to poorer countries [2], the unequal provision of vaccines has important consequences for wealthier countries and the international order [3–5]. The risks from low global provision include economic and health threats arising from continued interruptions of global supply chains, and the preservation of reservoirs that facilitate emergence of SARS-CoV-2 variants/mutations. One study estimates global costs at €9 trillion [6]. It has been estimated that approximately 70% of the worldwide population must be fully vaccinated to end the COVID-19 pandemic [7]. The delta variant has pushed the threshold for global herd immunity to 80% and potentially approaching 90% [8]. The rise of the Omicron variant cannot be stopped with the vaccines that are currently available, but at least the spread can be slowed down and morbidity and mortality associated with the disease can be reduced. Thus, vaccine inequity is not only a humanitarian disaster, it is one that has direct material consequences for wealthier countries.

Researchers at the International Monetary Fund (IMF) estimate the costs of global vaccination at $50 billion [6]. Other estimates put costs closer to €80 billion. Though there is no clear determination of what a fair share is, as a benchmark, if the richest 25% of countries provided €70 per citizen this would sum to 80 billion Euros and imply contributions of about €6 billion for Germany and €23 billion for the United States. Using the Fair Share calculation, based on OECD guidelines, Germany's share of OECD donor shares would be €5bn (8% of €63bn) (for details see [9]). Given its size and wealth this is about four times the average corresponding contribution of other wealthy nations.

However, a fair distribution of COVID-19 vaccines is not only a matter of providing money, as vaccines continue to be scarce for some time. Despite the importance of globally distributing COVID-19 vaccines to stop the pandemic, most Western countries have launched campaigns for a third shot and a number of countries even started the distribution of a fourth vaccination (e.g. Israel, USA, Chile, Denmark or Germany) in the wake of the Omicron wave [10]. Poorer countries will therefore continue to struggle to obtain sufficient vaccines for their citizens in the foreseeable future. On 4 August 2021, WHO director Tedros Adhanom Ghebreyesus therefore called for a moratorium halting COVID-19 vaccine boosters in favor of unvaccinated (see Reuters).

Given that governments need to secure public support either for making large monetary contributions or for sharing vaccines with poorer countries, it is important to understand the levels and drivers of public support for global vaccine sharing and to identify ways through which governments can increase solidarity with other countries in need. Previous research has focused on ethical questions of global vaccine distribution [11, 12] or on mapping the international distribution of COVID-19 vaccines [13, 14]. However, little is known about public opinion on global vaccine sharing. In this study we focus on three motivations: *intrinsic* motivations, *material* motivations, and *strategic* motivations. By intrinsic motivations we refer to the preferences for sharing that derive from concerns for the well-being of global populations. By material incentives we refer to the economic and health benefits to German citizens that might arise from a global response. By strategic incentives, we refer to considerations that make the preferred German contribution dependent on what other states are providing to global vaccine sharing. We hereby build on related literatures on contributions to global public goods. In particular, past work on European solidarity during the Eurozone crisis [15–18] and preferences for international climate agreements [19, 20], highlights how popular preferences

are not only affected by both the specific costs and benefits for the donor country, but also by the design of multilateral agreements and the behavior of other countries. Building on this work we examine both the role of (stipulated) direct costs to Germany and the structure of international cooperation. We also assess the extent to which public support can be increased through information campaigns appealing to the self-interest of citizens.

## Materials and methods

Our study draws on survey data and data from two survey experiments that we fielded in a multi-wave panel study in Germany using the online access panel of the survey company Respondi. Respondi relies on online channels and offline channels to recruit new panelists for its online panel. After completing a profiling questionnaire covering basic sociodemographic information, panelists are then invited to participate in surveys. Respondi compensates its panelists for completing a survey.

Our target population consists of all German citizens aged 18 to 75 years. In wave 1, the sample corresponded to the official national statistics with respect to age, sex and region, though the quality of this correspondence weakened somewhat by wave 4 (for details, see Table 1 and Section A in the S3 File). We conducted the experiments in wave 2 and 4 of the panel. All analyses were specified in preregistered analysis plans and the study obtained IRB

**Table 1. Sociodemographic characteristics.**

|  | Federal Statistical Office | Wave 1 | Wave 2 | Wave 4 (panel) | Wave 4 (refreshment) |
|---|---|---|---|---|---|
| Age |  |  |  |  |  |
| 18–29 | 0.18 | 0.19 | 0.14 | 0.10 | 0.18 |
| 30–39 | 0.18 | 0.17 | 0.16 | 0.15 | 0.17 |
| 40–49 | 0.16 | 0.18 | 0.18 | 0.19 | 0.18 |
| 50–59 | 0.22 | 0.22 | 0.24 | 0.26 | 0.23 |
| 60–75 | 0.26 | 0.24 | 0.28 | 0.30 | 0.25 |
| Sex |  |  |  |  |  |
| Female | 0.50 | 0.50 | 0.48 | 0.44 | 0.50 |
| Male | 0.50 | 0.50 | 0.52 | 0.56 | 0.49 |
| Other | 0.00 | 0.00 | 0.00 | 0.00 | 0.00 |
| Region |  |  |  |  |  |
| Baden-Württemberg | 0.13 | 0.13 | 0.13 | 0.12 | 0.13 |
| Bavaria | 0.16 | 0.16 | 0.16 | 0.16 | 0.16 |
| Berlin | 0.04 | 0.04 | 0.05 | 0.05 | 0.04 |
| Brandenburg | 0.03 | 0.03 | 0.03 | 0.03 | 0.03 |
| Bremen | 0.01 | 0.01 | 0.01 | 0.01 | 0.01 |
| Hamburg | 0.02 | 0.02 | 0.02 | 0.02 | 0.02 |
| Hesse | 0.08 | 0.08 | 0.08 | 0.07 | 0.07 |
| Mecklenburg-Vorpommern | 0.02 | 0.02 | 0.02 | 0.02 | 0.02 |
| Lower Saxony | 0.10 | 0.10 | 0.10 | 0.10 | 0.10 |
| North Rhine-Westphalia | 0.22 | 0.22 | 0.22 | 0.22 | 0.22 |
| Rhineland-Palatinate | 0.05 | 0.05 | 0.05 | 0.05 | 0.05 |
| Saarland | 0.01 | 0.01 | 0.01 | 0.01 | 0.01 |
| Saxony | 0.05 | 0.05 | 0.05 | 0.05 | 0.05 |
| Saxony-Anhalt | 0.03 | 0.03 | 0.03 | 0.03 | 0.03 |
| Schleswig-Holstein | 0.03 | 0.03 | 0.03 | 0.03 | 0.03 |
| Thuringia | 0.03 | 0.03 | 0.03 | 0.03 | 0.03 |

approval at Humboldt-Universität zu Berlin (HU-KSBF-EK 2021 0019). Data and replication material are publicly available at https://wzb-ipi.github.io/vaccine_solidarity/.

## Experiment 1

The first experiment draws on data generated from a factorial online survey experiment that was implemented in wave 4 of the panel study which was administered to 10,525 respondents between 8 and 22 September 2021. In the experiment, participants first received an introductory text explaining the need for global vaccine sharing to overcome the pandemic. Afterwards, respondents were asked to consider a hypothetical scenario which randomly varied along four dimensions (health benefits, economic benefits, number of countries participating, contribution of other countries). Respondents were then asked to indicate how much Germany should contribute to global vaccine sharing, both in € and in vaccine doses. Each respondent received two vignettes successively.

More specifically, two dimensions focused on the benefits of global vaccine sharing for Germany in terms of public health and in terms of economic growth. One asked participants to imagine that "The risk of new mutations of the coronavirus increases considerably in Germany if there are no vaccinations in poorer countries"; a second asked participants to imagine that "The German economy shrinks by around 5% if there are no vaccinations in poorer countries." For each of these a control condition was provided in which there were no costs to Germany if there are no vaccinations in poorer countries (see Table 2).

The other two dimensions focused on the nature of multilateral agreements, asking participants to imagine settings in which 0, 20 or 40 countries took part, contributing collectively €0, €20, or €40 billion. In all there are five types of agreement considered (because 0 participants implies 0 contributions and vice versa). In all this gives rise to a 2 × 2 × 5 factorial design (see complete wording of the factor levels in the appendix).

## Experiment 2

The second survey experiment was conducted in wave 2 of the panel study which was fielded between 29 April and 10 May 2021 (see Section D in the S3 File for further information). In this experiment, participants were randomly assigned to a *treatment group* that is exposed to a video explaining the benefits of global vaccine sharing and a *control group* which did not see the video. The video emphasizes in particular the risk of more mutations forming if vaccines are not made available in developing countries. Subsequently, respondents were asked whether they support the international distribution of vaccines (attitudinal outcome) and were offered the opportunity to donate money to UNICEF which was put in charge for global vaccine

**Table 2. Experimental design: Factors and levels.**

| Factor | Level |
| --- | --- |
| **Trade** | (0) **No negative impact** on German economy if no vaccinations in poorer countries |
| | (1) German economy **shrinks by 5%** if no vaccinations in poorer countries |
| **Risk** | (0) **Risk of new mutations does not increase** if no vaccinations in poorer countries |
| | (1) **Risk of new mutations increases** if no vaccinations in poorer countries |
| **Deal** | (0) **No international deal** on the distribution of vaccines to poorer countries. |
| | (1) International deal: **20 other countries, total 20 billion euros** |
| | (2) International deal: **40 other countries, total 20 billion euros** |
| | (3) International deal: **20 other countries, total 40 billion euros** |
| | (4) International deal: **40 other countries, total 40 billion euros** |

sharing (behavioural outcome). More specifically, respondents earned 75 so-called "Mingle Points" cents in token appreciation for their participation in the survey (this corresponds to 0.75 Euros). We offered them 50 additional Mingle Points and gave them the following choice. They could either keep the 50 Mingle Points for themselves or donate all or part of them to UNICEF for the worldwide distribution of Corona vaccines. For every point they donated, we donated 1.5 Euro Mingle points to UNICEF (see Table 4 in the S3 File).

We only present results for the personal donation outcome since the conjoint had a flawed design. For full transparency we present results of the conjoint experiment in the S3 File.

### Survey data on beneficiary prioritization

To measure humanitarian motivations directly, we included a question in both waves 2 and 4 asking respondents to indicate how the German government should prioritize to ensure vaccination for an older Indian woman as compared to a younger German woman in the second and fourth wave of the panel study.

### Analytic approach

We implement two types of analysis.

The first type of analysis using regression analysis to compare groups across conditions. For the primary analysis we estimate the effects of treatment conditions on two outcomes, *Cash*-contributions (in billion Euros), and *Doses* (millions). Estimates are generated by regressing outcomes on four conditions (*Amount given by other countries*, *Number of others giving*, *Risk*, *Trading importance*) along with all interactions and allowing for individual level fixed effects.

$$
\begin{aligned}
Y_{ij} = {} & \beta_0 + \beta_1 X_1 + \beta_2 X_2 + \beta_3 X_3 + \beta_4 X_4 + \\
& \beta_5 X_1 X_2 + \beta_6 X_1 X_3 + \beta_7 X_1 X_4 + \beta_8 X_2 X_3 + \beta_9 X_2 X_4 + \beta_{10} X_3 X_4 + \\
& \beta_{11} X_1 X_2 X_3 + \beta_{12} X_1 X_2 X_4 + \beta_{13} X_1 X_3 X_4 + \beta_{14} X_2 X_3 X_4 + \\
& v_i + \epsilon_{ij}
\end{aligned}
$$

All conditions ($X_1$—$X_4$) are centered on zero which allows us to read average effects directly from main terms ($\beta_1$—$\beta_4$) [21]. Estimates are generated together with robust standard errors using the `estimatr::lm_robust` function in R [22].

We use the same analytic procedure when analyzing the effects of the video treatment from Experiment 2 though here there is only one treatment and one response per respondent and there are no interaction terms or fixed effects included in the analysis.

A second type of analysis estimates parameters from a structural model. For this analysis we assume that individuals evaluate own country ($y_i$) and other country (($y_j$)) contributions according to the objective function:

$$
u = (\alpha + \beta \times \text{economic risk} + \delta \times \text{health risk}) \times \log(\sum\nolimits_{-i} y_j + y_i) - \gamma \times (y_i - \kappa \bar{y})^2 - y_i^2
$$

where $\bar{y}$ is the average contribution of other states, and $\sum_{-i} j_j$ is the total contributions by others. We note that this expression differs from the expression in our pre-analysis plan in not subscripting parameters by respondent. This model builds on standard models of a multi-player public goods production problem (see e.g. [23]). The basic form is $bg-c$ where $g$ is the public good produced, $b$ is marginal benefits and $c$ is the cost of contributions. We allow marginal benefits to depend on the type of public good specified by treatment variations (health risks and economic risks). We also assume diminishing returns to the public good and convex costs. We depart from classic models by allowing for the possibility of peer effects—that

subjects seek to benchmark their contributions to those of others (see e.g. [24]). Together $\gamma$ and $\kappa$ capture how much ($\gamma$) and how ($\kappa$) individuals might seek to match Germany's contributions to those of others, whether they value Germany giving more ($\kappa > 1$), less $\kappa < 1$ or the same ($\kappa = 1$) as the average contribution of other countries.

Letting $y_i^*$ denote optimal own contributions given economic and health risks and the contributions of others, we assume that players report $y_i^* + \epsilon_i$ where $\epsilon_i \sim N(0, \sigma^2)$ captures deviations from predictions of the model.

We estimate parameters ($\alpha, \beta, \delta, \gamma, \kappa, \sigma$) using maximum likelihood implemented via `bbmle::mle2` in R.

Finally our analysis of beneficiary prioritization uses simple means in respondents answers.

## Results

### Experiment 1

Fig 1 graphs raw data patterns to show the share of Germans supporting contributions of €$X$ or less (top panel) or $X$ donations of vaccine doses (bottom panel) or less. We indicate, separately, preferred contributions when there are high costs meaning that there are large economic and health costs of the status quo for Germany (versus low costs meaning there are no economic and health costs) and a situation of high multilateralism meaning that there is a major international deal (versus low multilateralism meaning there is no international deal).

We see that median favored contributions are around €2 bn and 100 million doses, an amount closely in line with actual current German commitments. There is considerable heterogeneity, however, with about one third support commitments around €5 bn or more. A small share—around 1 in 8, support much larger contributions. Given the skewed nature of

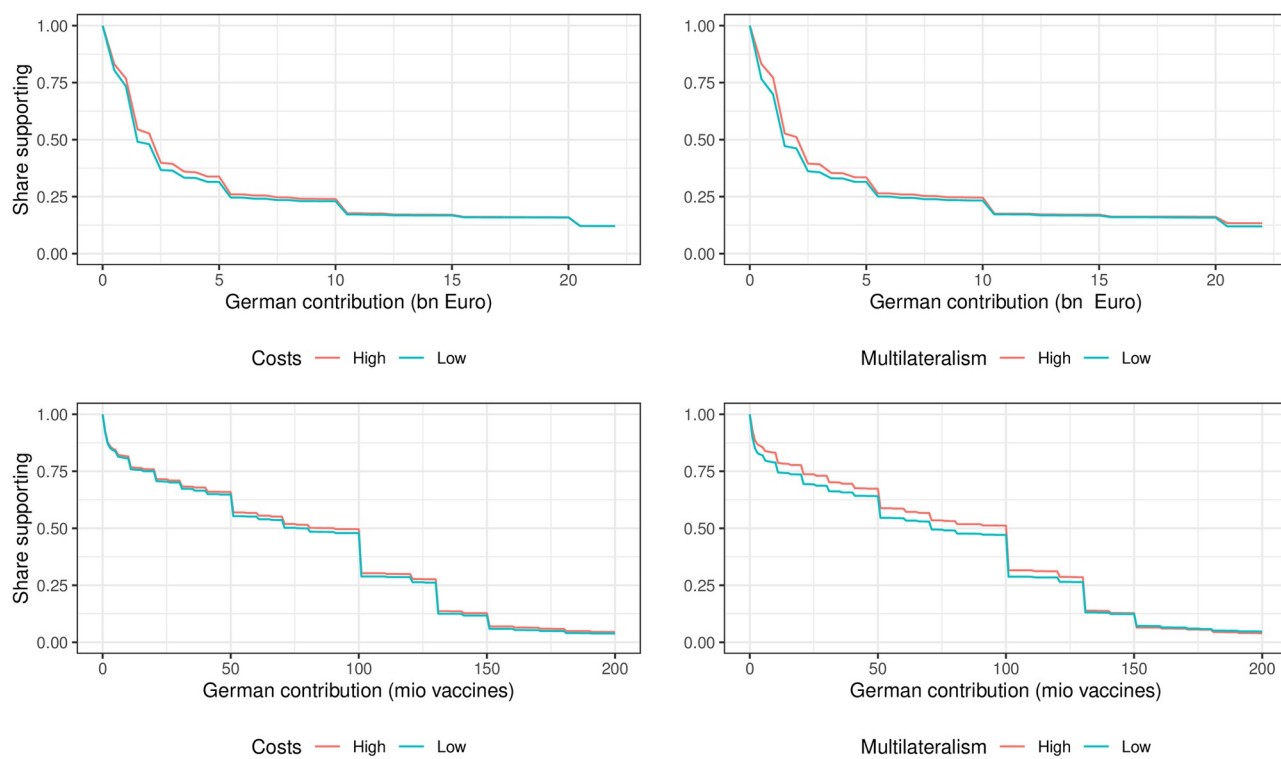

**Fig 1. Distribution of support for contributions of different sizes.**

the distributions, the average amount proposed is much larger—around €8 bn Euros for cash, but the mean number of doses proposed is lower at around 80 million.

In Figs 6–9 in Section E of the S3 File we document variation in these levels of support across subgroups. Based on preregistered analyses we show first, that support varies substantially as a function of political party support— with the greatest support among Green party voters and voters of the social democratic SPD whereas support is weakest among voters of the right-wing populist AfD. Second we show variation as a function of migration background, with substantially greater support for higher levels of solidarity among respondents with a migration background. We note nevertheless that despite this variation, support is high in all groups. To wit, a majority of AfD supporters, the group least likely to support international vaccine solidarity, still support contributions of 1 billion Euros or more.

In Fig 1 we can also see that both sets of conditions increase the size of contribution supported by survey participants. These differences are generally statistically significant (see below) but as seen from the raw data, the effects are quantitatively small.

Fig 2 shows the marginal effects of all conditions on optimal cash donations and vaccine doses. This represents the same underlying patterns as seen in Fig 1 though the focus here is on average effects. Overall we see that both health risks and trading importance are statistically distinguishable from zero. While this effect may be small compared to Germany's overall budget, it is a sizable effect when using the median donations as a benchmark.

Strikingly the amounts offered are positively responsive to *amounts* given by other countries, but unresponsive to the *number* of other countries giving. This is the opposite to what one might expect from accounts that focus on free-riding between states understood as the failure to contribute a fair share to global vaccine distribution if other states already donate large amounts. Our results instead suggest a willingness to support initiatives regardless of average contributions by other countries.

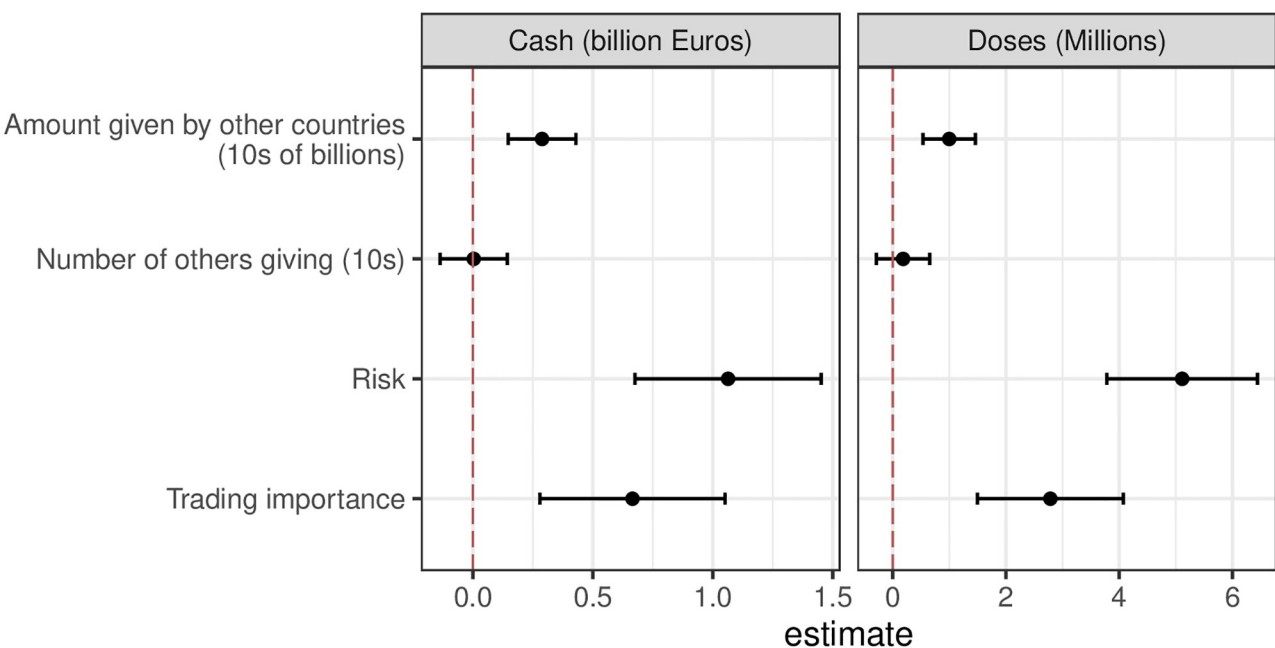

**Fig 2. Marginal effects of conditions.**

**Table 3. Structural parameter estimates.**

| parameter | estimate | std.error | statistic | p.value | conf.low | conf.high |
|---|---|---|---|---|---|---|
| $\alpha$ | 240.62 | 2.01 | 119.73 | 0.00 | 236.68 | 244.56 |
| $\beta$ | -10.32 | 12.71 | -0.81 | 0.42 | -35.23 | 14.59 |
| $\delta$ | 37.79 | 12.45 | 3.04 | 0.00 | 13.40 | 62.19 |
| $\gamma$ | 0.61 | 0.07 | 9.22 | 0.00 | 0.48 | 0.75 |
| $\kappa$ | 5.22 | 0.40 | 13.06 | 0.00 | 4.44 | 6.00 |
| $\sigma$ | 15.98 | 0.08 | 205.18 | 0.00 | 15.83 | 16.14 |

Our heterogeneity analysis (in appendix) suggest that these treatment effects are quite similar for respondents that support different parties or that have different migration backgrounds.

Our preregistered structural estimation lets us shed more light on the intrinsic, material, and strategic motivations of respondents. The estimated structural parameters from this analysis are given in Table 3.

We see here strong evidence for marginal gains from contributions independent of economic and health risks ($\alpha$), these marginal gains are increased ($\delta$) when there are substantial health risks (by about 16%), but are not much affected by economic risks ($\beta$). Respondents place weight on alignment with other countries ($\gamma$), but the results on $\kappa$ suggest that they target contributions that are significantly higher (five times higher) than the average amount given by other nations. Note that average contributions by other countries was not provided directly to respondents, though it can be calculated from the numbers and the amounts given. The implied factor of 5 is remarkably close—and somewhat higher—than the factor of 4 that we calculated using the fair share calculation.

## Experiment 2

Consistent with the findings presented above we found evidence for quite high baseline willingness to contribute in experiment 2 implemented in the second wave of the panel study. The median contribution is 40% and the most common response—chosen by about a third of respondents—was to contribute the full amount of this funding to UNICEF. Fig 3 reports the results of the information experiment. We find that respondents in the treatment group that were exposed to the video are significantly more likely to show solidarity both with regard to the attitudinal and the behavioral outcome. More specifically, the reported willingness to personally support international distribution of vaccines is 0.069 units higher than in the control group. In a similar vein, the average actual donations that respondents made to UNICEF were significantly higher in the treatment than in the control group. The average effect of treatment is a 4 percentage point increase in amounts offered.

## Priorities analysis

Turning to the analysis of prioritization, we find that 57% of the respondents place the priority of the older Indian woman as high or higher than the German woman, and 38% even place it strictly higher.

## Discussion

Our results show that German citizens are supportive of generous contributions to the global distribution of vaccines against COVID-19. Median popular support is somewhat below

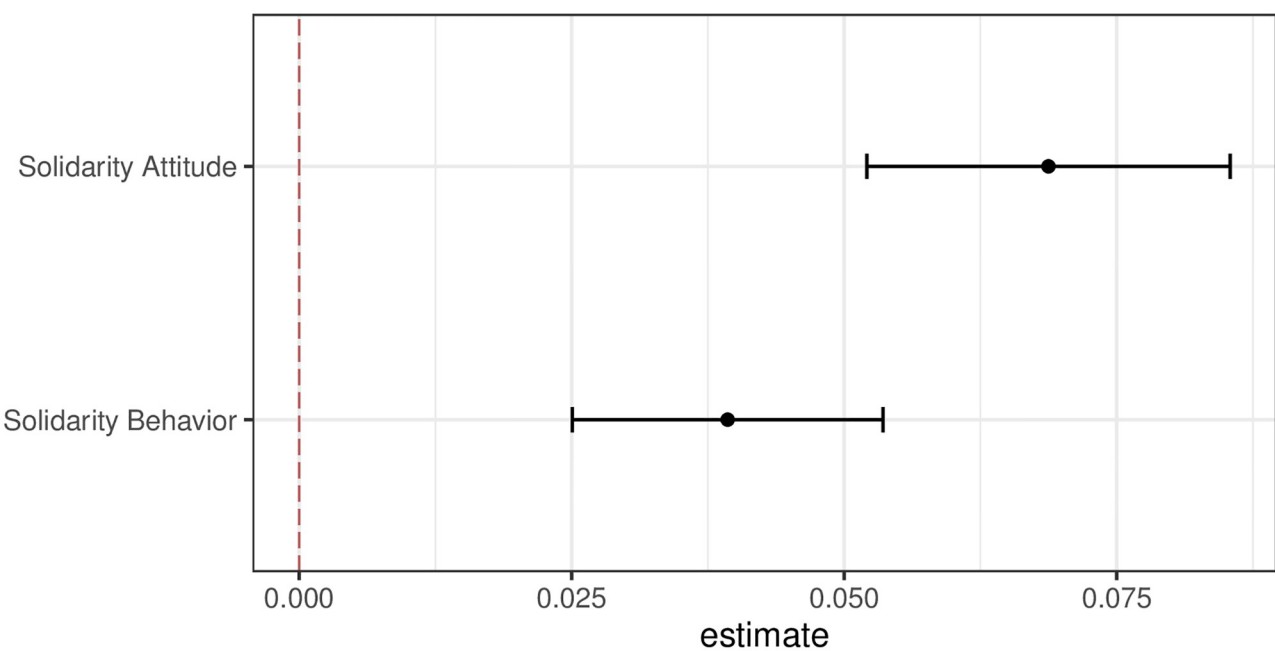

**Fig 3. Effect of video treatment on individual solidarity.**

estimated contributions needed to fund global vaccine redistribution; though average amounts exceed this. Popular preferences for global vaccine solidarity appear to primarily stem from humanitarian concerns and are only in part explained by material and strategic considerations. While there is a preference for multilateral efforts, public support for large contributions to global vaccine distribution does not depend on the behavior of other countries. We furthermore show that information campaigns can increase public support for international vaccine solidarity.

We interpret the high rates of solidarity in the control conditions—when there are no health or economic benefits stipulated—as evidence of non-strategic humanitarian concerns. Our analysis of support priorities supports this interpretation. The comparison we examined there directly pitted nationalist concerns against humanitarian concerns. We would expect any strategic considerations that enter in the decision to prioritize an Indian woman should apply *a fortiori* to the prioritization of a German woman.

Overall these results support the conclusions that German citizens see intrinsic benefits from global contributions, that these are augmented, but do not depend on externalities— health externalities in particular—and that Germans, insofar as they benchmark contributions to those of other nations, do not support free riding and indeed value contributing at rates higher than those of other wealthy nations. Our results moreover show that there is not only a high baseline willingness to contribute to global vaccine sharing, but that the solidarity expressed by citizens can be significantly increased by providing information about the benefits of global vaccine distribution.

Before concluding, we discuss a set of plausible threats to validity. The first relates to a set of critique in [25] that highlights difficulties that respondents have in providing numeric valuations of public goods. Focusing on contingent valuation surveys in economics, [25] highlight risks that respondents may not have well defined valuations of public goods, may not be in a position to take relevant budget constraints into account (which can lead to what is sometimes called an embedding effect), and provide answers that cannot easily be assessed against

revealed preferences. Although our experiment bears similarities with contingency valuation surveys our aim here is not to assess individual willingness to pay for global vaccinations, but rather to assess what policies respondents would like to see public officials follow. Their report of these preferences to us, which in turn get communicated publicly, albeit in aggregated form, bears a direct relation to the policy quantities of interest which is absent for contingency valuation surveys; by the same token, embedding effects may indeed be real in the formulation of policy priorities. However, this does not imply a bias in measurement, but rather a threat of inconsistencies in policy demands.

Although the aim of our study is different to those of scholars assessing the valuation of public goods, we still recognize that survey responses may not accurately predict the policy preferences that would advocate in a policy context when the benefits and costs of different strategies are debated by political actors, and responses may be sensitive to question wording, for instance, or the information we supplied to contextualize costs. Fully addressing these questions would require implementing a field analogue of our study, perhaps involving discussions between politicians and citizens [26].

Our findings have clear implications for the current debate on international vaccine solidarity in the COVID pandemic. With regard to the instruments of achieving a fair allocation of vaccines globally, the evidence presented in Figs 1 and 2 shows that patterns of support for monetary and dose donations are similar. However, the value of median preferred dosage donations (90 million doses) is substantively smaller than the median preferred cash donations (2 billion) if we assume the costs per vaccine to be around 6 Euro per dose [6]. This is in part an artifact of the fact that dosage donations by Germany are capped by vaccine holdings; nevertheless it highlights the fact that framing sharing in terms of vaccines rather than cash, by suggesting a zero sum nature of the problem, may yield weaker support for sharing. There are, however, many other avenues to address vaccine inequality including strategies that target production in developing countries, through financing, extending intellectual property rights, and sharing know-how.

We acknowledge that this study only uses stated preferences for hypothetical scenarios so that it remains unclear whether the effect sizes would be the same under similar real world scenarios. However, it has been shown that stated preferences are indeed predictive of actual behavior [27]. We only use data from an online survey in Germany and rely on an access panel. For future research it would be beneficial to use different modes and multiple other countries as well.

## Conclusions

Our study has shown that the stark inequalities in global vaccine distribution are not in line with public opinion. Vaccine nationalism, though evident in policy, is a minority position in Germany. The results of our study have important implications for the current public debate on global vaccine distribution, but also for international solidarity and international cooperation more generally. On the one hand, the COVID-19 vaccination is a highly salient issue for all citizens worldwide and thus provides a unique opportunity to study popular preferences for international solidarity. On the other hand, since herd immunity is a global public good as the pandemic can only be overcome if all countries worldwide are immunized, our findings can furthermore inform the literature on preferences for international cooperation.

Our study also has important implications for the debate on international solidarity [15–18] and global public goods [19, 20] more generally. We show that public support for international solidarity in global vaccine sharing is positively affected by the amounts given by other countries which corroborates the findings from previous research on global climate

cooperation and EU bailouts [15, 19]. However, there is no evidence for free-riding as citizen support is not conditional on the number of other countries participating in global vaccine sharing, in contrast to findings on public attitudes towards global climate agreements [19]. Finally, our study adds to the current debate by showing public support for international solidarity is malleable and that informing citizens about the benefits of vaccine sharing could increase support.

While efforts to share vaccines globally have been a failure to date, the evidence we provide importantly suggests that this is not due to lack of public support for the proposition. Average proposals exceed current contributions by the German government and also exceed fair share calculations of what the German government ought to be providing. *Median* proposals are somewhat less than the fair share benchmark, but nevertheless large. Evidence from our experiment with ancillary data suggest that strategic considerations matter, but are not paramount in explaining public preferences. Rather humanitarian rationales loom large. In all, these results suggest that policy-makers who take up the mantel of addressing the challenge of achieving global vaccination will have the moral support of the public behind them.

## Supporting information

**S1 File.**
(PDF)

**S2 File.**
(PDF)

**S3 File.**
(PDF)

## Author Contributions

**Conceptualization:** Ferdinand Geissler, Felix Hartmann, Heike Klüver, Johannes Giesecke.

**Formal analysis:** Ferdinand Geissler, Felix Hartmann, Macartan Humphreys.

**Funding acquisition:** Macartan Humphreys, Heike Klüver, Johannes Giesecke.

**Methodology:** Ferdinand Geissler, Felix Hartmann, Macartan Humphreys, Heike Klüver, Johannes Giesecke.

**Project administration:** Heike Klüver.

**Visualization:** Felix Hartmann, Macartan Humphreys.

**Writing – original draft:** Felix Hartmann, Macartan Humphreys, Heike Klüver.

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
