## [Decision Letter · Decision Letter 0]

20 Jun 2022

PONE-D-22-09640Public support for global vaccine sharing in the COVID-19 pandemic: Intrinsic, material, and strategic driversPLOS ONE

Dear Dr. Geissler,

Thank you for submitting your manuscript to PLOS ONE. After careful consideration, we feel that it has merit but does not fully meet PLOS ONE’s publication criteria as it currently stands. Therefore, we invite you to submit a revised version of the manuscript that addresses the points raised during the review process.

Both reviewers are pleased with the overall framework of the paper. They, however, suggested important revisions that need to be carefully addressed before further considering this manuscript for publication. 

We look forward to receiving your revised manuscript.

Kind regards,

Sara Rubinelli

Academic Editor

PLOS ONE

Journal Requirements:

Reviewers' comments:

Reviewer's Responses to Questions

**Comments to the Author**

1. Is the manuscript technically sound, and do the data support the conclusions?

Reviewer #1: Yes

Reviewer #2: Yes

2. Has the statistical analysis been performed appropriately and rigorously? 

Reviewer #1: I Don't Know

Reviewer #2: Yes

3. Have the authors made all data underlying the findings in their manuscript fully available?

Reviewer #1: No

Reviewer #2: Yes

4. Is the manuscript presented in an intelligible fashion and written in standard English?

Reviewer #1: Yes

Reviewer #2: No

5. Review Comments to the Author

Reviewer #1: The manuscript by Geissler et al. is interesting, relevant, and timely. However, the structure/organization of the manuscript is confusing (Results described in the Introduction, Methods described in the Results, Discussion of findings in Results, etc.). Moreover, several key pieces of information about the sample, procedures, and analytical approach are not described, limiting this manuscript’s public health contributions.

Specific comments go below:

• Title

Include location and time frame of the study in the Title.

• Abstract

What does top/bottom 10 mean? Largest economies? Human development index?

“we show that global inequities are out of line with domestic German public opinion.” Is editorialized. Please reframe.

“we see no evidence for free riding on contributions by other states.” Please explain further.

• Intro

First paragraph needs citations.

Please describe “strategic motivations” and how they differ from “intrinsic motivations” in greater detail.

The authors present findings and their implications in the Introduction, which is confusing. The 2 last paragraphs in the Introduction should be moved to the Results and Discussion, respectively.

• Methods

While the factorial experiment procedures are described in detail, more information on recruitment activities, inclusion/exclusion criteria, and analysis methods are needed in this section. For example: was this a nationally representative sample? Who was eligible to participate?

The first paragraph of the Structural analysis should be moved to the Methods.

More information on how the authors arrived at their structural model should be provided.

• Results

What are main sociodemographic characteristics of the sample? How do they map onto Germany’s national demographics?

In Figure 1, how did you define high vs. low costs and multilateralism?

What are thresholds for considering effect sizes small vs. large? A difference of over 1 billion Euros in investment and 50 million doses between high vs. low risk do not seem “not large”

In Figure 2, what does the treatment effect of “amount given by others” mean? For each 10 billion more Euros other countries invest, participants believed Germany should invest about 0.25 billion more?

Please provide further explanation on the meaning of parameters gamma, sigma, and kappa in your structural function.

Please provide your working definition of “free riding”.

The last paragraph of the Results should be discussed in the Discussion.

• Discussion

It is very hard to follow the second and third paragraphs of the Discussion. It is unclear how the second paragraph of the Discussion relates to this paper. Please explain. Were the experiments discussed in the second and third paragraphs of the Discussion conducted as part of the present study or were they separate?

It seems that new findings are presented in the second and third paragraphs of the Discussion. These should be moved to the Results section.

The last paragraphs of the discussion should also discuss Limitations in the sampling approach (was this a nationally representative sample? How were participants recruited? Phone interviews? Online? Etc.)

• Conclusion

The first paragraph of the Conclusion should be moved to the Discussion. How are the authors comparing monetary vs. vaccine contributions?

“Finally, our study adds to the current debate by showing that governments could increase support for international solidarity through information campaigns.” This point is overstated. It is not clear if treatment effects seen in your experiment would be reproduced in mass information campaigns.

Reviewer #2: 1.Title

OK

2.Abstract

The statement ……over 85 % vaccinated in the top 10 countries and below 3 % in the bottom 10. It is not clear who are top ten and bottom ten, what is the basis of classification is it GDP? World Bank ranking? Poverty index. It should be stated clear

3.Background/Introduction

• There are multiple currency being used in the paper USD$ and Euro € para. 3 page 2…. the costs of global vaccination at $50 billion… Since the text is predominantly reported in Euro, this should translated to the same currency

• The background aims at establishing the main question that the paper is trying to answer and link it with previous research in the field, and why it is important. The paper largely has managed to provide that. However, on. Pg. 2… the last paragraph …Our results show that German citizens are supportive of generous contributions to the global distribution of vaccines against COVID-19. Median popular support is somewhat below estimated contributions…. From this paragraph to the end of the introduction section the authors start presenting results, which is contrary to the aim of this section, this paragraphs will serve better by being moved into results, discussion and conclusion sections.

4.Methods

OK

5.Results

OK

6.Discussion

OK

7.Conclusion and Recommendation(s) Most of the text in para 1 and 2 pg. 6 on conclusion section, while providing valid information, the authors continue discussing the results instead of drawing a conclusion. It is important that authors ONLY conclude the study results and implications based on results and discussion. No need to use reference in this section, rather conclude the study.

8.References

OK

9.Figures

Need to improve the quality of the figures

6. PLOS authors have the option to publish the peer review history of their article (what does this mean?). If published, this will include your full peer review and any attached files.

Reviewer #1: No

Reviewer #2: **Yes: **George M Ruhago

---

## [Author Response · Author response to Decision Letter 0]

4 Aug 2022

Dear Reviewers,

Thank you very much for the thoughtful reviews of our manuscript.

We have endeavored to address all of your concerns and suggestions. We have thoroughly revised the manuscript on the basis of the reviewer and editor comments. 

Below, we explain in detail the modifications we have made. Editor and reviewer comments are written in italics; our responses follow in normal font. To indicate changes that we have conducted in the manuscript, we highlight all changes in the main text in red.

Comments to the Author

1. Is the manuscript technically sound, and do the data support the conclusions?

Reviewer #1: Yes

Reviewer #2: Yes

2. Has the statistical analysis been performed appropriately and rigorously?

Reviewer #1: I Don't Know

Reviewer #2: Yes

3. Have the authors made all data underlying the findings in their manuscript fully available?

Reviewer #1: No

Reviewer #2: Yes

● All our data is publicly available at https://wzb-ipi.github.io/vaccine_solidarity/. We have mentioned this before only in the first footnote, but we are now making this much more explicit by directly stating the availability of the data and replication materials in the main text of the research design section. 

4. Is the manuscript presented in an intelligible fashion and written in standard English?

Reviewer #1: Yes

Reviewer #2: No

● We carefully revised the entire paper to make sure that there are no typographical or grammatical errors. 

5. Review Comments to the Author

Reviewer #1: The manuscript by Geissler et al. is interesting, relevant, and timely. However, the structure/organization of the manuscript is confusing (Results described in the Introduction, Methods described in the Results, Discussion of findings in Results, etc.). Moreover, several key pieces of information about the sample, procedures, and analytical approach are not described, limiting this manuscript’s public health contributions.

Specific comments go below:

• Title

Include location and time frame of the study in the Title.

● We changed the title to “Public support for global vaccine sharing in the COVID-19 pandemic: Evidence from Germany” and clearly added the time frame to the abstract and the Methods discussion.

• Abstract

What does top/bottom 10 mean? Largest economies? Human development index?

● We changed the sentence as follows:

As of September 2021 an estimated 27% of the global population was fully vaccinated for COVID-19 but the global distribution of vaccines was extremely unequal, with 70% or more vaccinated in the ten countries with the highest vaccination rates and less than 1% in ten countries with the lowest vaccination rates.(FOOTNOTE Numbers calculated from figures from Our World in Data for countries with reported vaccinations and with populations of at least 500,000. See replication materials) 

“we show that global inequities are out of line with domestic German public opinion.” Is editorialized. Please reframe.

● We have reframed the sentence as follows:

“We demonstrate that the majority of German citizens are against global inequalities in vaccine distribution.”

“we see no evidence for free riding on contributions by other states.” Please explain further.

● We have reformulated the sentence as follows:

“in particular we see no evidence for free riding on contributions by other states as the level of preferred donations are positively responsive to the contributions of other countries.” 

• Intro

First paragraph needs citations.

● We have added a citation to Mathieu at al. (2021) on which basis we have calculated the reported numbers. The paragraph now reads as follows:

“Vaccination is the key to overcome the COVID-19 pandemic. Multiple vaccines have been developed in record time and more than 9 billion doses have been administered globally to date. As of early 2022 an estimated 50% of the global population is fully vaccinated. However, the global distribution of the COVID vaccines is extremely unequal (Mathieu et al. 2021): Fewer than 1% of doses administered have been administered in low income countries. While about 85% of citizens are vaccinated in the ten countries leading in the COVID vaccination campaign, less than 3% of the population is vaccinated in the ten least performing countries. Strikingly, per capita GDP alone explains as much as 60% of the variation in vaccination rates; COVID mortality about 6%

“The numbers are calculated from data from Our World in Data; calculations available in replication materials.”(Footnote: Numbers calculated based on data provided by Mathieu et al. (2021); calculations available in replication materials.)

Please describe “strategic motivations” and how they differ from “intrinsic motivations” in greater detail.

● To address this point, we have rewritten the text passage in question as follows:

“By strategic incentives, we refer to considerations that make the preferred German contribution dependent on what other states are providing to global vaccine sharing. We hereby build on related literatures on contributions to global public goods. In particular, past work on European solidarity during the Eurozone crisis (Bechtel, Hainmueller and Margalit, 2017; Kuhn, Nicoli and Vandenbroucke, 2020; Kuhn, Solaz and van Elsas, 2018; Stoeckel and Kuhn, 2018) and preferences for international climate agreements (Bechtel, Genovese and Scheve, 2019; Bechtel and Scheve, 2013), highlights how popular preferences are not only aﬀected by both the speciﬁc costs and beneﬁts for the donor country, but also by the design of multilateral agreements and the behaviour of other countries. Building on this work we examine both the role of (stipulated) direct costs to Germany and the structure of international cooperation.”

The authors present findings and their implications in the Introduction, which is confusing. The 2 last paragraphs in the Introduction should be moved to the Results and Discussion, respectively.

● Following the suggestions of the reviewer, we have moved the last two paragraphs to the Results and Discussion sections. 

• Methods

While the factorial experiment procedures are described in detail, more information on recruitment activities, inclusion/exclusion criteria, and analysis methods are needed in this section. For example: was this a nationally representative sample? Who was eligible to participate?

● In order to address this comment, we have added a new Section in the Supplementary Material (Section A) and have carefully revised the methods section and extended the text passages in question as follows:

“Our study draws on two survey experiments that we fielded in a multi-wave panel study in Germany. At the start of the panel, the sample was nationally representative according to age, sex and region (for details, see the supplementary materials). We conducted the experiments in wave 2 and 4 of the panel. In order to conduct the factorial survey experiment, we fielded the experiment relying on the online access panel of the survey company Respondi. Respondi relies on online channels and offline channels to recruit new panelists for its online panel. After completing a proling questionnaire covering basic sociodemographic information, panelists are then invited to participate in surveys. Respondi compensates its panelists for completing a survey. Our population of interest consists of all German citizens aged 18 to 75 years.”

The first paragraph of the Structural analysis should be moved to the Methods.

● We thank the reviewer for this suggestion. Having carefully examined the paper again, we worried that this material would be important for readers to understand the analysis. Instead of moving this part we have tried to make this whole discussion more readable. 

More information on how the authors arrived at their structural model should be provided.

We have added the following motivation to the text:

● “This model builds on standard models of a multiplayer public goods production problem (see e.g. Hellwig (2013)). The basic form is bg - c where g is the public good produced, b is marginal benefits and c is the cost of contributions. We allow marginal benefits to depend on the type of public good specified by treatment variations. We also assume diminishing returns to the public good and convex costs. We depart from classic models by allowing for the possibility of peer effects — that subjects seek to benchmark their contributions to those of others (see e.g. Fischbacher and Gächter (2010).” 

replacing the previous overly brief account:

“The function assumes diminishing marginal returns to total contributions, convex costs of own contributions, substitutability of own and other contributions for global benefits, and a possible desire to benchmark contributions against the contributions of others.” 

• Results

What are main sociodemographic characteristics of the sample? How do they map onto Germany’s national demographics?

● We added a new table with sociodemographic characteristics comparing official data with the distribution in the different waves in the Supplementary Material (Table 3). We have also added a new Section in the Supplementary Material (Section B) to show which impact panel attrition has on the results of Experiment 1. 

In Figure 1, how did you define high vs. low costs and multilateralism?

● We did two things to make it easier to understand these terms. First, we added Table 1 to show the factors and levels of the experimental design. Second, we reformulated the text passage as follows:

“We indicate, separately, preferred contributions when there are high costs meaning that there are large economic and health costs of the status quo for Germany (versus low costs meaning there are no economic and health costs) and a situation of high multilateralism meaning that there is a major international deal (versus low multilateralism meaning there is no international deal).”

What are thresholds for considering effect sizes small vs. large? A difference of over 1 billion Euros in investment and 50 million doses between high vs. low risk do not seem “not large”

● We have reformulated the sentence to give a benchmark for the effect sizes.

“Overall we see that both health risks and trading importance are statistically distinguishable from zero, but, as seen already, the magnitude of effects is not large considering that the federal budget of Germany for 2021 was around € 500 bn.”

In Figure 2, what does the treatment effect of “amount given by others” mean? For each 10 billion more Euros other countries invest, participants believed Germany should invest about 0.25 billion more?

● Yes, the interpretation is correct. In order to make it clearer to the readers, we changed the label in Figure 2 from “amount given by others” to “amount given by other countries”.

Please provide further explanation on the meaning of parameters gamma, sigma, and kappa in your structural function.

● gamma: gamma captures the weight that players place on the extent to which their contributions are in line with those of other states 

● kappa:kappa captures the extent to which players, if they benchmark their contributions to those of others, seek to match (kappa = 1), exceed (kappa >1) or provide less than (kappa <1) others

● sigma: captures the variance of an error term that allows for deviation from the predictions of the mode; if the model explains behavior very tightly, sigma will be small; 

We now provide more explanation for these terms in the text. 

Please provide your working definition of “free riding”.

● We have added the following discussion to the main text:

“This is the opposite to what one might expect from accounts that focus on free-riding between states understood as the failure to contribute a fair share to global vaccine distribution if other states already donate large amounts. Our results instead suggest a willingness to support initiatives regardless of average contributions by other countries.”

The last paragraph of the Results should be discussed in the Discussion.

● As suggested we moved the last paragraph of the Results Section to the Discussion Section.

• Discussion

It is very hard to follow the second and third paragraphs of the Discussion. It is unclear how the second paragraph of the Discussion relates to this paper. Please explain. Were the experiments discussed in the second and third paragraphs of the Discussion conducted as part of the present study or were they separate? It seems that new findings are presented in the second and third paragraphs of the Discussion. These should be moved to the Results section.

● We agree that it was somewhat confusing how we presented the results of our two experiments in the original version of the manuscript. In order to address this point, we have therefore carefully revised the entire methods, results and discussion section to explain in detail that we conducted two different survey experiments as part of wave 2 and wave 4 of our panel survey in Germany.

The last paragraphs of the discussion should also discuss Limitations in the sampling approach (was this a nationally representative sample? How were participants recruited? Phone interviews? Online? Etc.)

● As suggested above, we have explained in much more detail how the sample is constructed and how respondents were recruited in the Materials and Methods. We also added a paragraph on limitations at the end of the Discussion Section.

“We acknowledge that this study only uses stated preferences for hypothetical scenarios so that it remains unclear whether the effect sizes would be the same under similar real world scenarios. However, it has been shown that stated preferences are indeed predictive of actual behavior (Hainmueller, Hangartner and Yamamoto, 2015). Moreover, we only use data from an online survey in Germany and rely on participants of an access panel. For future research it would be beneficial to use different modes and multiple other countries as well.”

• Conclusion

The first paragraph of the Conclusion should be moved to the Discussion. 

● As suggested, we have moved the paragraph to the Discussion Section.

How are the authors comparing monetary vs. vaccine contributions?

● Because the scales of the two different measures are distinct we do not directly compare the treatment effects. When comparing the median of the preferred dosage donation and the preferred cash donation, we now explain that we follow the estimated costs per dosage by Agarwal and Gopinath (2021). We therefore reframed the conclusion as follows:

“However, the value of median preferred dosage donations (90 million doses) is substantively smaller than the median preferred cash donations (2 billion) if we assume the costs per vaccine to be around 6 Euro per dose (Agarwal and Gopinath, 2021).”

“Finally, our study adds to the current debate by showing that governments could increase support for international solidarity through information campaigns.” This point is overstated. It is not clear if treatment effects seen in your experiment would be reproduced in mass information campaigns.

● We have reframed the sentence as follows:

“Finally, our study adds to the current debate by showing public support for international solidarity is malleable and that informing citizens about the benefits of vaccine sharing could increase support.”

Reviewer #2: 

1.Title

OK

2.Abstract

The statement ……over 85 % vaccinated in the top 10 countries and below 3 % in the bottom 10. It is not clear who are top ten and bottom ten, what is the basis of classification is it GDP? World Bank ranking? Poverty index. It should be stated clear

● We changed the sentence as follows:

“As of early 2022 an estimated 50% of the global population is fully vaccinated for COVID-19 but the global distribution of vaccines is extremely unequal, over 85% vaccinated in the top 10 countries and below 3% in the 10 countries with the lowest vaccination rate”

3.Background/Introduction

• There are multiple currency being used in the paper USD$ and Euro € para. 3 page 2…. the costs of global vaccination at $50 billion… Since the text is predominantly reported in Euro, this should translated to the same currency

● We have converged all US-Dollar indications to Euros.

• The background aims at establishing the main question that the paper is trying to answer and link it with previous research in the field, and why it is important. The paper largely has managed to provide that. However, on. Pg. 2… the last paragraph …Our results show that German citizens are supportive of generous contributions to the global distribution of vaccines against COVID-19. Median popular support is somewhat below estimated contributions…. From this paragraph to the end of the introduction section the authors start presenting results, which is contrary to the aim of this section, this paragraphs will serve better by being moved into results, discussion and conclusion sections.

● In order to address this point, we have moved the paragraph in question to the Discussion Section.

4.Methods

OK

5.Results

OK

6.Discussion

OK

7.Conclusion and Recommendation(s) Most of the text in para 1 and 2 pg. 6 on conclusion section, while providing valid information, the authors continue discussing the results instead of drawing a conclusion. It is important that authors ONLY conclude the study results and implications based on results and discussion. No need to use reference in this section, rather conclude the study.

● In order to address this point, we have moved all discussions of results to the Discussion Section and only discuss broader implications in the Conclusion.

8.References

OK

9.Figures

Need to improve the quality of the figures

● This seems to be an issue with the manuscript where the figures are presented in reduced quality. All figures were uploaded to PLOS One in high quality and can be downloaded using the links in the appendix of the manuscript (e.g., Figure 1: https://www.editorialmanager.com/pone/download.aspx?id=30923249&guid=28f0b479-470b-4b23-a0c1-e710710dd939&scheme=1).

---

## [Decision Letter · Decision Letter 1]

24 Aug 2022

PONE-D-22-09640R1Public support for global vaccine sharing in the COVID-19 pandemic: Evidence from GermanyPLOS ONE

Dear Dr. Geissler,

Thank you for submitting your manuscript to PLOS ONE. After careful consideration, we feel that it has merit but does not fully meet PLOS ONE’s publication criteria as it currently stands. Therefore, we invite you to submit a revised version of the manuscript that addresses the points raised during the review process.

Both reviewers are pleased with the revision. Yet, one reviewers still has minor comments that I kindly ask you to address before I can make a final decision. 

We look forward to receiving your revised manuscript.

Kind regards,

Sara Rubinelli

Academic Editor

PLOS ONE

Journal Requirements:

Reviewers' comments:

Reviewer's Responses to Questions

**Comments to the Author**

1. If the authors have adequately addressed your comments raised in a previous round of review and you feel that this manuscript is now acceptable for publication, you may indicate that here to bypass the “Comments to the Author” section, enter your conflict of interest statement in the “Confidential to Editor” section, and submit your "Accept" recommendation.

Reviewer #1: (No Response)

Reviewer #2: All comments have been addressed

2. Is the manuscript technically sound, and do the data support the conclusions?

Reviewer #1: Yes

Reviewer #2: Yes

3. Has the statistical analysis been performed appropriately and rigorously? 

Reviewer #1: Yes

Reviewer #2: Yes

4. Have the authors made all data underlying the findings in their manuscript fully available?

Reviewer #1: Yes

Reviewer #2: Yes

5. Is the manuscript presented in an intelligible fashion and written in standard English?

Reviewer #1: Yes

Reviewer #2: Yes

6. Review Comments to the Author

Reviewer #1: The authors have been largely responsive to my initial comments. A few minor comments remain:

1- Abstract:

* Information in the abstract does not match the Introduction; please update the abstract (50% of the global population is vaccinated, etc., 85% vaccination in top 10 countries, etc.).

2- Introduction:

* Factual data needs citation (number of doses administered globally, etc.).

* I suggest you change "disease mutation" to "emergence of SARS-CoV-2 variants/mutations".

* Vaccines are not highly effective in preventing transmission of Omicron variants, but they still prevent morbidity and mortality associated with the disease. This should be noted in the 2nd paragraph.

* Several countries have recommended a fourth dose for some populations, not only Israel (US, Brazil, etc.). Please update.

3- Methods

* I believe there is a confusion regarding the meaning of "nationally representative sample". Having the same proportion of men/women, or the same age distribution of the general population does not necessarily mean that the sample is nationally representative. Representativeness of the sample is more about the sampling strategy and if everyone in the target population could have been in the study. Is that the case in this study? Is this a random sample of the German population? Is this a systematic sampling approach?

* Data access: I could not access the data file through the weblink provided, only the code used in analyses. Perhaps I am missing something obvious here, but I suspect other readers may also have trouble accessing the data used in this report.

* If possible, move Supplemental Table A to the main text, as we know that many readers do not access suppl materials.

*There should be a section in the Methods that describes the analytical approach employed.

4- Results

* I still think that the specification of the models should be discussed/explained in the Methods.

* I disagree that effect sizes are small. Take, for example, the effect of risk, which leads to a billion euro increase in donations (50% of the median contribution). This may be small compared to Germany's overall budget, but it seems like a substantial effect using the median donations as a benchmark. The responsiveness to health risks is also aligned with the significant effect of the educational video in increasing donations in Experiment 2.

* It is unclear to me why authors used the dichotomous outcome (donate all to UNICEF vs. not) rather than the continuous amount of Mingle Points donated. Please clarify.

5- Discussion

* The authors continue to present new findings in the Discussion. These should be moved to the Results (and included in the Methods) or excluded from this report.

Page 6: "we furthermore included a question asking respondents to indicate how the German government should prioritize to ensure vaccination for an older Indian woman as compared to a younger German woman in the second and fourth wave of the panel study.The comparison directly pits nationalist concerns against humanitarian concerns. Importantly also, we expect, any strategic considerations that enter in the decision to prioritize an Indian woman should apply a fortiori to the prioritization of a German woman. Strikingly, we nd that 57% of the respondents place the priority of the older Indian woman as high or higher than the German woman, and 38% even place it strictly higher."

* Were participants made aware of the cost of vaccines? The mismatch between money vs. vaccine contributions may be due to participants not knowing the average cost of vaccine doses.

Reviewer #2: (No Response)

7. PLOS authors have the option to publish the peer review history of their article (what does this mean?). If published, this will include your full peer review and any attached files.

Reviewer #1: No

Reviewer #2: **Yes: **GEORGE MUGAMBAGE RUHAGO

---

## [Author Response · Author response to Decision Letter 1]

12 Oct 2022

Reviewer #1: The authors have been largely responsive to my initial comments. A few minor comments remain:

1- Abstract:

* Information in the abstract does not match the Introduction; please update the abstract (50% of the global population is vaccinated, etc., 85% vaccination in top 10 countries, etc.).

• We thank the reviewer for pointing to this inconsistency. We have updated all numbers to match the timing of the survey. Though this is not the most recent data it is the most relevant for interpreting results and avoids working with a moving target.

2- Introduction:

* Factual data needs citation (number of doses administered globally, etc.).

• Thank you. We write now in a footnote: 

“Numbers calculated based on data provided by Mathieu (2021); calculations available in replication materials. Top 10 and bottom 10 countries are from among 102 countries with populations > 500,000 that report vaccination rates.”

• We also now more clearly indicate the relevant section of the replication file (it is now labeled “3. Numbers provided in Introduction (Vaccine background data from OWID)”) and all calculations are implemented more transparently with consistent dates.

* I suggest you change "disease mutation" to "emergence of SARS-CoV-2 variants/mutations".

• We agree that this is more accurate and have changed it accordingly.

* Vaccines are not highly effective in preventing transmission of Omicron variants, but they still prevent morbidity and mortality associated with the disease. This should be noted in the 2nd paragraph.

• We reformulated the sentence as follows to capture this as well:

“The rise of the Omicron variant cannot be stopped with the vaccines that are currently available, but at least the spread can be slowed down and morbidity and mortality associated with the disease can be reduced..”

* Several countries have recommended a fourth dose for some populations, not only Israel (US, Brazil, etc.). Please update.

• That is correct. In the meantime more countries followed with this recommendation. We have adjusted this in the text.

“Despite the importance of globally distributing COVID-19 vaccines to stop the pandemic, most Western countries have launched campaigns for a third shot and a number of countries even started the distribution of a fourth vaccination (e.g. Israel, USA, Chile, Denmark or Germany) in the wake of the Omicron wave (Mahase, 2021).”

3- Methods

* I believe there is a confusion regarding the meaning of "nationally representative sample". Having the same proportion of men/women, or the same age distribution of the general population does not necessarily mean that the sample is nationally representative. Representativeness of the sample is more about the sampling strategy and if everyone in the target population could have been in the study. Is that the case in this study? Is this a random sample of the German population? Is this a systematic sampling approach?

• We agree with the reviewer. We have reformulated this part making it even more clear that the sample is not a "nationally representative sample", though we also detail how on available observable characteristics of the sample match the population. We also reformulated the paragraph in the Supplementary Materials A. accordingly

“Our study draws on two survey experiments that we fielded in a multi-wave panel study in Germany using the online access panel of the survey company Respondi. Respondi relies on online channels and offline channels to recruit new panelists for its online panel. After completing a profiling questionnaire covering basic sociodemographic information, panelists are then invited to participate in surveys. Respondi compensates its panelists for completing a survey.

Our target population consists of all German citizens aged 18 to 75 years. In wave 1, the sample corresponded closely to the official national statistics with respect to age, sex and region though the quality of this correspondence weakened somewhat by wave 4 (for details, see Table 1 and Section A in the Supplementary Materials). We conducted the experiments in wave 2 and 4 of the panel. All analyses were specified in preregistered analysis plans and the study obtained IRB approval at [Redacted Institution]. Data and replication material are publicly available at https://wzb-ipi.github.io/vaccine_solidarity/”.

* Data access: I could not access the data file through the weblink provided, only the code used in analyses. Perhaps I am missing something obvious here, but I suspect other readers may also have trouble accessing the data used in this report.

• The data will be made available under the provided weblink as soon as the paper gets published.

• We have edited the replication code however so that it runs from the cleaned data which we have made available already. As of now the reviewer should be able to replicate all analyses from the github replication archive. 

* If possible, move Supplemental Table A to the main text, as we know that many readers do not access suppl materials.

• Following this suggestion, we moved the Table into the main text.

*There should be a section in the Methods that describes the analytical approach employed.

• We have adopted this suggestion and added a new section on analytic procedures/

4- Results

* I still think that the specification of the models should be discussed/explained in the Methods.

• We have adopted this suggestion and added a new section on analytic procedures/

* I disagree that effect sizes are small. Take, for example, the effect of risk, which leads to a billion euro increase in donations (50% of the median contribution). This may be small compared to Germany's overall budget, but it seems like a substantial effect using the median donations as a benchmark. The responsiveness to health risks is also aligned with the significant effect of the educational video in increasing donations in Experiment 2.

• We agree that the size of the effects could be evaluated differently depending on the benchmark. Therefore we reformulated the sentence.

“Overall we see that both health risks and trading importance are statistically distinguishable from zero. While this effect may be small compared to Germany's overall budget, it is a sizable effect when using the median donations as a benchmark. 

* It is unclear to me why authors used the dichotomous outcome (donate all to UNICEF vs. not) rather than the continuous amount of Mingle Points donated. Please clarify.

• We do not use the dichotomous outcome in the analysis, we use the full data. In fact much of the data is in the all or nothing categories (see replication material section 8). This note on the differences in the highest category was intended to provide intuition for the movement behind the estimate which largely reflects this increase in the highest category. We have removed this point however to avoid confusion and simply note the average effect. 

5- Discussion

* The authors continue to present new findings in the Discussion. These should be moved to the Results (and included in the Methods) or excluded from this report.

Page 6: "we furthermore included a question asking respondents to indicate how the German government should prioritize to ensure vaccination for an older Indian woman as compared to a younger German woman in the second and fourth wave of the panel study.The comparison directly pits nationalist concerns against humanitarian concerns. Importantly also, we expect, any strategic considerations that enter in the decision to prioritize an Indian woman should apply a fortiori to the prioritization of a German woman. Strikingly, we nd that 57% of the respondents place the priority of the older Indian woman as high or higher than the German woman, and 38% even place it strictly higher."

• We moved this paragraph to the Results section.

* Were participants made aware of the cost of vaccines? The mismatch between money vs. vaccine contributions may be due to participants not knowing the average cost of vaccine doses.

• We did not tell the respondents directly about the average costs of vaccine doses. However, we stated that around 11 billion vaccine doses would be needed globally and that the costs to cover these would be around 70 billion (see Figure 5). So the respondents had an idea about the actual costs of one vaccine dose.

Reviewer #2: (No Response)

---

## [Decision Letter · Decision Letter 2]

15 Nov 2022

Public support for global vaccine sharing in the COVID-19 pandemic: Evidence from Germany

PONE-D-22-09640R2

Dear Dr. Geissler,

We’re pleased to inform you that your manuscript has been judged scientifically suitable for publication and will be formally accepted for publication once it meets all outstanding technical requirements.

Kind regards,

Sara Rubinelli

Academic Editor

PLOS ONE
---

## [Editor Report · Acceptance letter]

21 Nov 2022

PONE-D-22-09640R2 

Public support for global vaccine sharing in the COVID-19 pandemic: Evidence from Germany 

Dear Dr. Geissler:

I'm pleased to inform you that your manuscript has been deemed suitable for publication in PLOS ONE. Congratulations! Your manuscript is now with our production department. 

Kind regards, 

on behalf of

Dr. Sara Rubinelli 

Academic Editor

PLOS ONE